# In Situ Synthesis of Silver Nanoparticles on Flame-Retardant Cotton Textiles Treated with Biological Phytic Acid and Antibacterial Activity

**DOI:** 10.3390/ma15072537

**Published:** 2022-03-30

**Authors:** Qingqing Zhou, Jiayi Chen, Zhenqian Lu, Qiang Tian, Jianzhong Shao

**Affiliations:** 1School of Materials and Textiles, Zhejiang Sci-Tech University, Hangzhou 310018, China; zhouqq0516@163.com; 2Yancheng Institute of Technology, College of Textiles and Clothing, Yancheng 224051, China; jychen1982@163.com (J.C.); luzhenqian2003@126.com (Z.L.); 3Zibo Dayang Flame Retardant Products Co., Ltd., Zibo 255000, China; 18953325077@189.com

**Keywords:** antibacterial treatment, phytic acid, flame-retardant cotton textiles, Ag NPs

## Abstract

Fabrics were flame-retardant finished using phytic acid, a cost-effective, ecologically acceptable, and easily available flame-retardant finishing chemical. Then, on the surface of the completed fabric, silver nanoparticles (Ag NPs) were grown in situ to minimize Ag NPs aggregation and heterogeneous post-finishing and to increase washing durability. Thus, flame-retardant and antibacterial qualities were added to textiles. The as-prepared textiles were evaluated for their combustion performance, thermal performance, and antibacterial capabilities. At the same time, their microstructures were studied using X-ray diffractometry (XRD), scanning electron microscopy (SEM), and X-ray photoelectron spectroscopy (XPS). The findings indicated that flame-retardant textiles had an excellent launderability (limiting oxygen index = 31% after 20 washing cycles). Meanwhile, Ag NPs-loaded flame-retardant textiles demonstrated self-extinguishing properties, with a limiting oxygen index (LOI) of 27%. Bacteriostatic widths of flame-retardant antibacterial textiles against *Escherichia coli* and *Staphylococcus aureus* were 5.28 and 4.32 mm, respectively, indicating that Ag NPs-loaded flame-retardant fabrics have certain flame-retardant and antibacterial capabilities. SEM and TEM analysis indicated that nanoparticles were uniformly dispersed over Ag NPs-loaded flame-retardant textiles and were around 20 nm in size. When compared to flame-retardant textiles, Ag NPs-loaded flame-retardant fabrics showed varied binding energy of P and N on the surface and Ag ion emergence. Thermogravimetric analysis at various heating rates revealed that the main pyrolysis temperature range of flame-retardant fabrics decreased, while the main pyrolysis temperature range of Ag NPs-loaded flame-retardant fabrics increased; the heating rate influenced the pyrolysis range but not the fabric mass loss. In situ reduction synthesis of Ag NPs-loaded flame-retardant textiles may successfully reduce agglomeration and heterogeneous dispersion of nano-materials during post-finishing.

## 1. Introduction

With the development of society, natural fibers and their products are widely used in all aspects of mankind. Although they decorate the living environment and improve the quality of life, their flammability has caused great trouble for human use. If textiles and clothing are in direct contact with human beings, once burned, they may cause partial skin burns, or large-area burns, which are life-threatening. Therefore, most textiles must meet certain flame retardant requirements to ensure the safety of the public and property. At present, the research on flame retardant technology of textiles, the development of various environmentally friendly flame retardant finishing agents and flame retardant textiles, and the formulation of laws and regulations on flame retardant textiles have become important research topics. In daily wear, clothing itself breeds bacteria due to sweat stains. At the same time, it is also inevitable to come in contact with external bacteria, which will cause certain harm to human health, because most pathogenic bacterial strains are increasingly resistant to conventional drugs, and global climate change supports the outbreak of these infections [1]. It is worth noting that pathogenic bacterial strains can spread infection from person to person through direct or indirect contact [2]. Antibacterial textiles gradually entered people’s lives and have become one of the most popular functional textiles in today’s market. Natural cotton fiber is welcomed by consumers because of its strong moisture absorption, soft hand feeling, comfortable wearing and other excellent wearing properties, but its flammable defects have also become the focus of attention in the textile field. Meanwhile, natural cotton fabric is considered to be a good growth substrate for microbial growth because it provides enhanced moisture, heat and serves as a carrier for the transmission of severe bacterial infections to humans [3]. Therefore, the preparation of flame retardant and antibacterial multifunctional textiles came into being. In order to improve the safety of people’s life and property, a fire and health barrier was built.

Phytic acid (PA), also known as inositol hexaphosphate, is an organic phosphorus additive with rich raw materials, friendly for the environment and has good biocompatibility. Phytic acid is widely used as an antioxidant, preservative and flame retardant in food, medicine, metal processing, textiles and other industries because of its high phosphorus content (28.16%) and strong chelation with metal ions [4,5]. Grunlan [6] applied phytic acid and chitosan to the flame retardant finishing of cotton fabric through layer-by-layer self-assembly technology for the first time. Chen’s research group at Suzhou University [7] used sodium phytate and chitosan as assembly units to carry out flame retardant finishing on silk fabrics. When 20 layers are assembled, the limiting oxygen index is 30. After 20 times of washing, the limiting oxygen index of silk fabric is still about 27%, which can be extinguished by itself in case of fire. At the same time, Tang’s research group also successfully applied phytic acid to polylactic acid knitted fabrics [8], wool [9] and silk [10]. Due to the strong acidity of phytic acid and the nature of alkali and acid resistance of cotton fabrics, the application of phytic acid is mainly focused on the flame retardant finishing of protein fiber, while the flame retardant finishing of cotton fabric is less.

Silver nanoparticles lead to the collapse of the membrane proton gradient and the destruction of many mechanisms of cell metabolism, resulting in cell death. Indeed, silver nanoparticles (Ag NPs) have been frequently employed as an antibacterial agent in the production of antibacterial textiles. Physical (e.g., physical crushing, vacuum condensation, mechanical milling) and chemical approaches have recently been used to synthesize Ag NPs (e.g., reduction, hydrothermal synthesis, gas-liquid dual-phase method, sol-gel method) [11,12,13]. Green reduction is one of these strategies that has received much attention. Plant-derived polysaccharides (e.g., maltodextrin and guar gum) can be employed as a reducing agent and stabilizer in the synthesis of Ag NPs [14,15]. Logeswari [16] et al. used five plant extracts to synthesize Ag NPs of varying sizes, and these Ag NPs showed outstanding antibacterial capabilities against harmful bacteria. Cotton textiles treated with Ag NPs, on the other hand, are restricted by heterogeneous dispersion and low laundry durability. As a result, in situ development of Ag NPs on fabric surfaces has become a popular study area. The Ag NP-loaded cotton textiles were created by the in situ reduction of Ag ions adsorbed on cotton fabrics with aloe vera extract. The Ag NPs were homogeneously dispersed on the surface and showed outstanding antibacterial efficacy against *Escherichia coli* (*E. coli,* Shanghai Bioresource Collection Center, Shanghai, China) and *Staphylococcus aureus* (*S. aureus,* Shanghai Bioresource Collection Center, Shanghai, China) [17]. Li et al. observed in situ reduction of Ag ions adsorbed on cotton fibers at room temperature by glow discharge [18]. The Ag NPs had an average size of 46 nm, and the produced Ag NPs-loaded textiles had good antibacterial action against *Bacillus subtilis* and *E. coli* even after 20 washing cycles. Wu et al. demonstrated cotton fabric grafting using methyl acrylamide and maleic anhydride-modified methyl acrylamide, followed by in situ reductions. After 50 washing cycles, the suggested grafted textiles showed an antibacterial rate of more than 97% against *E. coli* and *S. aureus* [19,20]. Zhang et al. created active cotton textiles with functionalities, such as decreasing Ag ions, regulating Ag NP size, and anchoring Ag NPs by oxidation and grafting using modified polyamide hyperbranched polymers. The size of the Ag NPs on the surface was 5~25 nm, and the produced cotton textiles had an antibacterial rate of 99.9% against *S. aureus* and *E. coli*. Furthermore, the cotton materials developed demonstrated great washing fastness [21].

Cotton textiles were grafted with PA using crosslinking EH, and then under the direction of a PA ligand soft template, Ag NPs were grown in situ on the fabric surface to produce multi-functional textiles with flame-retardant and antibacterial capabilities, providing a unique method for multi-functional finishing of cotton fabrics.

## 2. Experimental

### 2.1. Materials

Phytic acid (PA, 70 wt%), polyethyleneimine (PEI, 10,000 Mw), sodium citrate and silver nitrate (AgNO_3_) were all supplied by Aladdin Reagent Co., Ltd. (Shanghai, China), Peptone, beef paste and agar were purchased by Sinopharm Chemical Reagent Co., Ltd. (Shanghai, China). The bleached cotton fabric (weight: 120 ± 10 g/m^2^, twill) and cross-linking agent (EH, 45%) were obtained from the local market.

### 2.2. Flame-Retardant Finishing

Samples were treated at 45 °C for 1 h with PEI and EH concentrations of 10%, rolled (80% liquid rate), and then dried at 80 °C and baked for 2 min at 120 °C. Subsequently, the samples were immersed in 5% EH solution, padded, and immersed in a 0.4 mol/L PA solution with a pH of 2 for 30 min before being immersed and padded again. The samples had an 80% liquid rate, and then were dried at 80 °C and baked for 2 min at 120 °C. Finally, samples were thoroughly cleaned (until pH = 7) with distilled water and dried at 80 °C.

### 2.3. Flame-Retardant Antibacterial Finishing

According to the references [22] and improvements are as follows, for 1 h, flame retardant textiles were submerged in a 0.1 mol/L silver nitrate solution at room temperature (bath ratio of 1:30). The textiles were then immersed in a 20 g/L sodium citrate solution and treated for 1 h at 90 °C. Finally, they were removed, washed and dried. Textiles with in situ Ag NP development were designated as PPA-C, whereas unfinished and flame-retardant fabrics were designated as O-C and PP-C, respectively.

### 2.4. Characterization

X-ray diffraction (XRD) was completed on D8 Advance, (Brooke, Ettlingen, Germany). The light tube voltage was 40 kV, the tube current was 40 Ma, the step size was 0.02°, and the test speed was 0.1 s/step.

SEM images were taken using a Nova Nano SEM 450 scanning electron microscope (FEI, Hillsboro, OR, USA). The surfaces of the fabrics were plated with gold before observations.

The PPA-C fabrics were sonicated at the power of 60 W for 2 h at room temperature, and the released Ag NPs were then characterized by HRTEM (JEM2100PLUS, JEOL, Tokyo, Japan) with point resolution ≤ 0.23 nm, line resolution = 0.14 nm, LaB6 crystal electron gun, acceleration voltage = 200 kV.

X-ray photoelectron spectroscopy (XPS) was performed with a thermo X-ray source system (ESCALAB 250XI, Thermo Fisher Scientific, Waltham, MA, USA) using a mono Al Ka X-ray source (1486.6 eV) at a reduced power of 100 w. The residual pressure in the analysis chamber ranged from 10^−9^ to 10^−10^ Pa and the photo-emitted electrons were collected at a take-off angle of 90°.

Thermogravimetry spectra analysis (TG) was conducted on a thermogravimetric analyzer (-209F3, NETZSCH-Gerätebau GmbH, Selb, Germany) under the air atmosphere at a heating rate of 5, 10, 15, 20 °C/min, and the temperature ranged from 20 °C to 700 °C. Samples were cut into power and placed in an aluminum oxide pan. About 10.0 mg of the sample was used in each case.

### 2.5. Assessment of Cotton Fabrics

Antibacterial activity was determined using the AATCC 147-2004 Antibacterial Activity Assessment of Textile Materials: Parallel Streak Method. The 15 mL sterilized nutritional agar medium was poured into the Petri dish to allow the agar to form gel before inoculation. Transfer 1.0 ± 0.1 mL of 24 h liquid culture medium to the test tube containing 1.0 ± 0.1 mL of distilled water with a pipette gun, and stir appropriately to mix evenly; Inoculate a strip-shaped agar on the surface of a strip-shaped agar plate with a dilution of about 4 mm, and transfer it to the surface of a strip-shaped agar plate; blank cotton fabric (50 × 25 mm^2^) and Ag NP loaded cotton fabric (50 × 25 mm^2^) was pressed into the agar surface in a pattern perpendicular to the inoculation line and stored at 37 °C for 24 h. The sample and agar media were kept in contact throughout the culture phase. The bacteriostatic zone is defined as the boundary between agar medium and a sample that does not support bacterial proliferation. The total width (*T*) of the sample and the bacteriostatic zone (*W*) were investigated, and the bacteriostatic zone of fabric was calculated by:*W* = (*T* − *D*)/2(1)
where *W* is the bacteriostatic zone, *T* is the total width and *D* is the sample width.

The flame retardancy of cotton fabric was investigated according to the ASTM D6413-08 standard method by the limiting oxygen index (LOI, the minimum volume percentage of oxygen in a mixture of oxygen and nitrogen) using an oxygen index measuring instrument (HC-2, Air Times, Beijing, China). The 10 cm × 5 cm fabrics were prepared and placed in a cylinder through which a mixture of nitrogen and oxygen was passed.

The uniformity of fabric in-situ synthesis is indirectly expressed by the K/S of the fabric surface. The K/S test was performed on the fabric surface using the CE7000A color measuring and matching equipment. After equipment calibration, the textiles were folded into four layers and evaluated.

## 3. Results and Discussion

### 3.1. Mechanism of In Situ Reduction

Each PA molecule has six non-coplanar phosphate linkages and PA chelated metal ions. PA may chelate with ions, such as Ag^+^, Cu^2+^, Fe^2+^, and Zn^2+^ in a matter of seconds upon contact with these ions, resulting in products with many complex rings and good stability. Indeed, these compounds remain stable in very acidic or alkaline environments. When flame-retardant materials are submerged in silver nitrate solution, the PA on the fabric becomes complexed with the Ag^+^ in the solution and adheres uniformly and securely to the fabric surface. Ag ions were reduced in situ to Ag NPs wrapped in PA in the presence of a soft template and a reducing agent. According to Feng et al., PA does not often produce a 1:1 compound when complexed with Ag ions [23]. Rather, it produces molecules with a multivalent structure. As a consequence, the binding properties of Ag NP-loaded textiles generated by in situ reductions under the direction of soft template PA are excellent.

Fabric finishing and the mechanism of reduction are shown in Figure 1. The reduction of Ag^+^ is by sodium citrate (adding a reducing agent) and the amino groups in PEI molecules. The reducibility of sodium citrate results from its unusual active methylene structure. The electronegative impact of oxygen atoms on two neighboring carbonyls causes the electron cloud density of the hydrogen atom on methylene (-CH_2_-) to fall dramatically and approach the protonation state. As a result, the H atom has a high degree of activity and the propensity to lose electrons. Under specific circumstances, a redox reaction between citrate and Ag^+^ may be detected, resulting in acetone-1,3-dicarboxylic acid, CO_2_, and Ag NPs (Figure 1b [22]. The N atoms on the amino groups of PEI molecules possess lone-pair electrons and exhibit a high affinity for Ag^+^ complexation. Meanwhile, lone-pair electrons may be used to generate an electrical source for Ag^+^ reduction. Primary, secondary, and tertiary amino groups are reducible (primary amino group > secondary and tertiary amino group). The reaction process is depicted in Figure 1c [24,25].

### 3.2. XRD Analysis

The crystal diffraction peaks of cellulose Ⅰ were typically reported at 2θ *=* 14.8° (101), 16.6° (101), 22.7° (002) and 34.6° (040). Figure 2 illustrates the XRD spectra of textiles prior to and the following finishing. As seen, the front location and form remained essentially unaltered, showing that the crystal area of textiles had minor damage during functional finishing. According to the literature, Ag NPs at 2θ *=* 38.2°, 44.3°, 64.4°, 77.5° were extremely compatible with the locations of the crystal planes (111), (200), (220), and (311) corresponding to PDFWIN#87-0597, showing that the Ag NPs had a face-centered cubic (FCC) structure [26,27]. However, Ag NPs generated in situ on textiles are tiny and there is no obvious absorption peak in Figure 2 (PPA-C). This is expected, since very little Ag is present in the sample compared to the cellulose content, and it is nanostructured, which further decreases the XRD signal. The presence of Ag is quite clear from the SEM and XPS [18]. It is similar to recent research, which used polyvinylpyrrolidone as a capping agent and N, N-dimethylacetamide as a reducing agent to create Ag NPs in a one-bath method [28]. Additionally, the diffraction peaks of Ag NPs were predicted to be 38.2°, 45.5°, 50.98°, and 72.49° [29].

### 3.3. SEM and TEM Analyses

Figure 3 A–F illustrates the surface morphology of textiles before and during finishing and combustion. Figure 3A,B illustrates photos magnified 5000 times, Figure 3C illustrates an image magnified 10,000 times, Figure 3E illustrates an image magnified 40,000 times, and Figure 3D,F illustrate images magnified 20,000 times. The longitudinal morphology of unfinished cotton textiles before burning is seen in Figure 3A. As noticed, its surface is relatively smooth, flat, and banded, with a tiny natural twist; fine and light ash is produced after burning. As seen in Figure 3B, its fibers are essentially broken.

On the other hand, following flame-retardant finishing, the surface of textiles is coated with a film, and inter-fiber adhesion is seen, indicating that PEI-PA has been applied to the fabric surface. Following burning, the cloth mainly retains its fiber form. Meanwhile, uniform bubbles were noticed on the fabric surface, resulting in a foam coke layer, as illustrated in Figure 3D. This is primarily due to developing an expanded flame-retardant system composed of PEI-PA and fibers. Here, PA serves as the acid supply, PEI serves as the gas source, and cellulose serves as the carbon supplier. The flame-retardant property of cellulose is mostly due to the porous foam coke layer on the surface of PA-PEI, which also acts as a heat insulator, an oxygen barrier, and a smoke suppressant. This layer impeded heat penetration through the condensed phase, oxygen entry into the combustion zone, and overflow of gaseous or liquid products created during fiber surface degradation.

Compared to unfinished and flame-retardant textiles, in situ reduced Ag NP-loaded flame-retardant fabrics displayed uniformly dispersed nanoparticles (small sizes), as seen in Figure 3E. Following burning, both bubbles of various sizes and nanoparticles were found on the Ag NP-loaded flame-retardant textiles, indicating that the Ag NP-loaded flame-retardant fabrics were generated in situ and the nanoparticles were dispersed uniformly over the fabric surface.

Sonication of flame-retardant textiles loaded with Ag NPs in ethanol for 2 h resulted in the formation of Ag NPs solution, which were studied by TEM. Figure 3G,H illustrates TEM images of Ag NPs of 50 and 20 nm diameters, respectively. Ag NPs have a rather consistent dimension (D) of 20 nm. Figure 3H’s zoom-in figure depicts a single Ag NP. The grain lattice fringes are crucial because they include regular and irregular bright and dark fringes. The regular fringe is spaced at 0.238 nm, which corresponds to the crystal plane (111) of Ag NPs (FCC). The regularity and irregularity in the formation of Ag NPs can be attributed to stacking defects generated by interactions between atoms in the PA matrix and Ag atoms [30].

### 3.4. XPS Analysis

The XPS spectra of textiles before and after finishing are shown in Figure 4A–D. The XPS spectra of unfinished flame-retardant and Ag NP-loaded flame-retardant textiles are shown in Figure 4A–D) depicting the elemental speciation of Ag, P and N. In comparison to unfinished and flame-retardant textiles, Ag is present on Ag NP-loaded flame-retardant fabrics in addition to C, O, N, and P, showing successful preparation of Ag NP-loaded flame-retardant fabrics.

The binding energy of Ag NPs generated using various processes may differ somewhat. The binding energies of Ag NPs have been reported to be 368.1 and 374.2 eV, 367.6 and 373.8 eV, respectively [31,32]. In this investigation, four binding energies of Ag NPs were detected in the Ag NP-loaded flame-retardant textiles, as shown in Figure 4B-Ag3d. In AgCl, 373.6 and 367.6 eV correspond to Ag3d3/2 and Ag3d5/2, respectively, whereas 373.1 and 367.2 eV belong to Ag^+^3d3/2 and Ag^+^3d5/2 [33,34]. This is consistent with earlier research and implies that Ag NPs were effectively generated on the surface of flame-retardant textiles by in situ reductions, with a trace of AgCl also detected. This is due to the presence of Cl^−^ in the distilled water utilized. Cl^−^ and Ag^+^ in the solution react during in situ reductions, resulting in precipitate on the fabric surface.

The binding energy of P on the surface of flame-retardant textiles is 134.0 and 133.5 eV under neutral circumstances. P binding energy reduces by 0.3 eV after complexation with Ag, and new peaks are found at 132.5 and 132.0 eV. During the complexation of PA with Ag^+^, O forms a strong π bonding between O atoms and the metal ions. Meanwhile, P contributes to producing massive bonds, causing major changes in P’s chemical environment and binding energy. Furthermore, Na is found on the fabric surface, resulting in a reduction in the binding energy of P. XPS spectra of flame-retardant textiles are shown in Figure 4D. Three binding energies have been identified, including 401.5 eV for -N-, 400.9 eV for -NH-, and 399.1 eV for -NH_2_, indicating the existence of amino groups matching the polyethyleneimine structure in flame-retardant textiles [35,36]. N on the surface alters after in situ reductions of Ag NPs by flame-retardant textiles, and a new peak at 399.7 eV is present. This might be related to the high complexing capacity of amino nitrogen in PEI to Ag^+^ due to the presence of lone-pair electrons. Indeed, lone-pair electrons serve as an electron source for Ag^+^ reduction. The binding energy of electrons in the inner layer of N increases due to this oxidation, and the binding energy is proportional to the electrical loss in oxidation. Furthermore, as compared to secondary and tertiary amino groups, the primary amino group has greater complexation and reduction capabilities. Only the binding energy of -NH_2_, which is involved in oxidation, rises to 399.7 eV as a result [37,38].

### 3.5. LOI Analysis

Cotton textiles are extremely flammable, with an LOI of 18%. The LOI of finished fabrics is shown in Table 1. Following the application of PEI/PA finishing, an expansion system comprised of PEI, PA, and cellulose was created, resulting in better flame-retardant qualities of the fabric. Even after 20 washing cycles, the LOI of the cloth climbed to 37%, and self-extinguishing was observed (LOI = 31%). Ag NPs were decreased in situ using PA as the soft film plate based on flame-retardant textiles. The textiles were thus endowed with flame-retardant and antimicrobial qualities. The as-prepared Ag NP-loaded flame-retardant textiles are self-extinguishing, with an LOI of 27%. According to the standard of flame retardant woven fabrics (GB 17591-1998), flame retardant fabrics are divided into two grades: (1) grade B1 when LOI ≥ 32% or damage length ≤ 15 cm, *t*_1_ and *t*_2_ ≤ 5S; (2) when LOI ≥ 26% or damage length ≤ 20 cm, *t*_1_ ≤ 15 s, *t*_2_ ≤ 10 s, it is grade B2(continuation time *t*_1_ and ignition time *t*_2_). The lower LOI might be attributable to two things. First, the one-hour heat treatment during in situ fabric reduction resulted in decreased fastness between PEI/PA and fibers, resulting in a decrease in flame-retardant efficacy. Second, coordination bonds between O atoms in -OH and metal ions were formed during the complexation of PA and Ag ions, resulting in less -OH in PA and higher concentrations of derivatives, such as phosphoric acid produced by combustion of Ag NP-loaded flame-retardant fabrics, resulting in a decreased flame-retardant performance of fabrics.

### 3.6. Homogeneity Analysis

Apparent depth is a term that relates to the perceptual depth that is reflected in the color of opaque solid materials. Typically, the perceived depth of textiles is stated in terms of the Kubela–Munk (K/S) ratio. The concentration of colorful compounds and darkness, in particular, is proportional to the K/S value [39]. K/S testing was used to assess the homogeneity of Ag NPs decreased in situ on fabric surfaces. The K/S curves of the as-prepared textiles are shown in Figure 5. The color depth difference (△K/S%) is obtained by subtracting the maximum from the minimum value, which is used to express the homogeneity of the fabric. The brighter the △K/S value, the better the levelness. △K/S is 0.189 after calculation which shows uniformity. The test was performed 20 times, and the standard deviation (*σ*), defined as the arithmetic square root of the arithmetic mean (i.e., variance) of the square of the departure from the mean, was computed. In probability statistics, the standard deviation is frequently employed to quantify the degree of statistical dispersion. The formula is:(2)$$\sigma =\sqrt{\frac{\sum_{i=1}^{n}{\left(x_{i}-\bar{x}\right)}^{2}}{n}}$$

In this investigation, σ was determined to be 0.05936, showing a highly homogenous distribution of Ag NPs over the fabric surface.

### 3.7. Antibacterial Properties

The antibacterial capabilities of textiles were evaluated using *E. coli* and *S. aureus*, which are typical Gram-negative and Gram-positive bacteria, respectively. Figure 6 illustrates the acquired results. Unfinished cloth appears to have a little antibacterial effect against *E. coli* or *S. aureus*, since germs were seen around the fabric (Figure 6a1,a2). The in situ reduced Ag NP-loaded flame-retardant textiles exhibit excellent bacteriostatic activity against *E. coli* and *S. aureus* (see Figure 6b1,b2 and Table 2). Bacteriostatic widths of flame-retardant textiles against *E. coli* and *S. aureus* were 5.28 and 4.32 mm, respectively, indicating the Ag NP-loaded flame-retardant bacteriostatic impact fabrics against *E. coli* is greater than that against *S. aureus*, similar to earlier findings. After 10 washing cycles, the bacteriostatic width of flame-retardant textiles containing Ag NP fell to 4.69 and 3.87 mm, respectively, for *E. coli* and *S. aureus*.

If the sterile colony grows in the area where the sample is in contact with agar, it is considered qualitatively that the sample has bacteriostasis. The wider the width is, the better the bacteriostasis of the sample is. Therefore, the sample has certain bacteriostasis.

Recent research has focused on the antibacterial mechanism of Ag NPs, emphasizing the release of Ag ions from Ag NPs [40]. The bacteriostatic action of Ag NPs is considered to be mostly due to Ag ions. Specifically, in water or wet air, the surface of Ag NPs oxidizes, releasing Ag ions. Ag ions are oxidants because they are positively charged cations formed when Ag atoms lose electrons. Cheon et al. asserted that the antibacterial activity of Ag NPs is mostly determined by their shape and is inversely proportional to their total surface area and the amount of Ag ions discharged [41]. According to some reports [42], the order of Ag ion release is connected to the shape of Ag NPs. The order and amount of Ag ions released are as follows: Ag nanospheres (34 µg/mL) > Ag nano-rods (32 µg/mL) > triangular Ag NPs (26 µg/mL) > hexagonal Ag NPs (15 µg/mL). Silver ions have been shown to interact with proteins (particularly react with the negatively charged thiol groups) and phospholipids associated with the proton pump of bacterial membranes. This leads to the collapse of the membrane proton gradient and the destruction of many mechanisms of cell metabolism, resulting in cell death. Silver ions obviously do not have a single mode of action. They interact with various bacteria in microorganisms, resulting in a series of effects from inhibition of growth, loss of infectivity to cell death. The mechanism depends on the concentration of silver ions and the sensitivity of microbial species to silver [43,44,45].

### 3.8. Thermal Performances

Generally, cellulose has a high heat resistance and is stable below 100 °C. As cellulose is heated over 140 °C, the remaining glucose group undergoes dehydration, resulting in chemical reactions, such as a reduction in the degree of polymerization and an increase in the carbonyl and carboxyl groups. When the temperature rises over 180 °C, the rate of cellulose cleavage increases progressively. When the temperature of the cellulose structure exceeds 250 °C, the glycosidic linkages and certain C-O and C-C bonds are broken, resulting in new products and low molecular weight volatile chemicals. When the temperature hits 370 °C, 40–60% of the fiber mass is lost, resulting in damage to the crystalline zone and a lower degree of polymerization; when the temperature approaches 400 °C, the remaining cellulose is aromatized and carbonized, forming graphite [46]. As seen in Figure 7, the deterioration of cotton textiles in air may be split into two stages: the primary stage of breakdown in the low-temperature zone and the residual oxidation stage. Pyrolysis occurs mostly between 280 and 380 °C, at which point 70% of the cellulose is broken into tiny molecular components with the precipitation of volatile gas. The flame-retardant treatment’s objective is to delay the development of flammable molecules during the first decomposition stage. The maximum heat release rate temperature (T_max_) of cloth steadily increases as the heating rate increases. At 380–550 °C, the residue is oxidized; at this temperature, the products of the first step are further oxidized and degraded, forming a carbon layer.

The curves PP-C and PPA-C in Figure 7 are thermogravimetric analysis (TGA) curves of flame-retardant textiles and flame-retardant fabrics loaded with Ag NP in the air at various heating rates. According to the O-C TGA curve, the initial decomposition temperature (T_i_) of flame-retardant textiles decreased from 255 to 190 °C, and the primary pyrolysis temperature range is shifted to 190–280 °C. T_max_ is lowered from 340 to 270 °C at a heating rate of 20 °C/min, and thermal weight loss at the main stage is reduced to 40%, demonstrating that PA functions as a flame retardant by inhibiting fabric burning. Meanwhile, the oxidation stage of the residue is increased to 700 °C in order to minimize the pace of pyrolysis. As the pyrolysis temperature range and T_max_ of Ag NP-loaded flame-retardant textiles rise, T_max_ increases from 273 to 297 °C when heated at a rate of 20 °C/min, and the thermal loss rate at the main stage stays unchanged when compared to flame-retardant fabrics. This might be because, during the in-situ development of Ag NPs, Ag ions form complexes with the hydroxyl groups in PA, resulting in a decreased concentration of phosphoric acid produced during PA burning. As a result, PA flame retardant properties deteriorate, and its maximum thermal breakdown rate temperature rises but stays lower than that of unfinished fabric.

As demonstrated by the DTG curves of unfinished, flame-retardant, and Ag NP-loaded flame-retardant fabrics, flame-retardant and unfinished fabrics degraded slightly at 550 °C, which may be attributed to continuous oxidation of the fabrics. In contrast, Ag NP-loaded flame-retardant fabrics remained relatively flat in this temperature range, indicating complete combustion of Ag NP-loaded flame-retardant fabrics in the early stage. Meanwhile, Ag NPs can prevent clothes from coming into contact with the environment, allowing for continued oxidation of materials. The heating rate affects the pyrolysis range but not the fabric mass loss. This is mostly due to changes in the time required to achieve the final pyrolysis temperature when heating rates are varied, resulting in varying reaction extents and pyrolysis temperatures. The pyrolysis reaction extent is lowered at high heating rates, and temperature differences and gradients are formed within and outside the sample, resulting in increased thermal hysteresis. The final pyrolysis temperature eventually rises [47]. TGA-related fabric characteristics are shown in Table 3.

In order to further compare and analyze the pyrolysis behavior of the three fabrics, Figure 8 shows TG-DTG at a 10 °C/min heating rate. It is obvious from the TG curve in Figure 8 that the pyrolysis behavior of the finished fabrics has changed from four stages to three, and the content of carbon residue at 700 °C increases, indicating that the flame retardation is conducive to hindering the pyrolysis and oxidation processes of the fabric. It can be seen from the DTG curve in Figure 8 that the maximum thermal decomposition rate temperature of the unfinished fabric is 329 °C, and that of the flame retardant fabric and flame retardant antibacterial fabric is 265 °C and 291 °C, respectively. The decrease in the maximum thermal decomposition rate temperature is mainly due to the decomposition of phytic acid into phosphoric acid and other derivatives at a certain temperature, the fabric is easy to decompose under acidic conditions.

## 4. Conclusions

Bio-PA was used as the flame-retardant agent in the fabrication of flame-retardant textiles. The textiles were then grown in situ with Ag NPs to provide them with flame-retardant and antibacterial characteristics. Both flame-retardant fabrics and Ag NP-loaded flame-retardant fabrics demonstrated self-extinguishing properties and acceptable laundering durability; bacteriostatic widths of flame-retardant antibacterial fabrics on *E. coli* and *S. aureus* were 5.28 and 4.32 mm, respectively, indicating that the bacteriostatic efficacy of Ag NP-loaded flame-retardant fabrics on *E. coli* is superior to that of *S. aureus*. Furthermore, nanoparticles with diameters of around 20 nm were uniformly dispersed on Ag NP-loaded flame-retardant textiles. In comparison to flame-retardant textiles, Ag NP-loaded flame-retardant fabrics displayed changing binding energy of P and N on the surface, as well as Ag emergence, suggesting Ag NP adherence to fibers. TGA at different heating rates indicated that the major pyrolysis temperature range of flame-retardant textiles dropped, while the main pyrolysis temperature range of Ag NP-loaded flame-retardant fabrics rose; the heating rate had an influence on the pyrolysis range but not on fabric mass loss.

## Figures and Tables

**Figure 1 materials-15-02537-f001:**
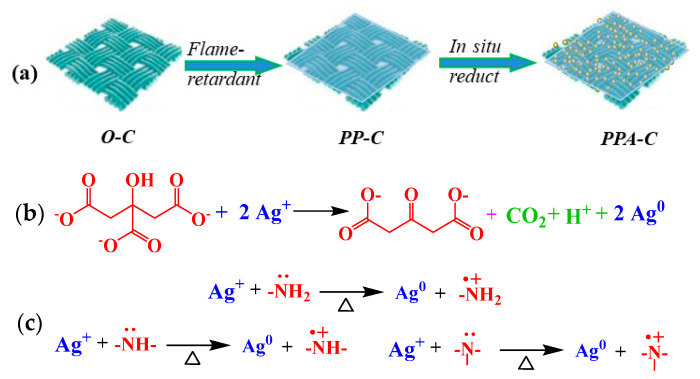
Schematic diagram of fabric finishing (**a**) and the reduction mechanism of sodium citrate (**b**) and amino groups in PEI molecules (**c**).

**Figure 2 materials-15-02537-f002:**
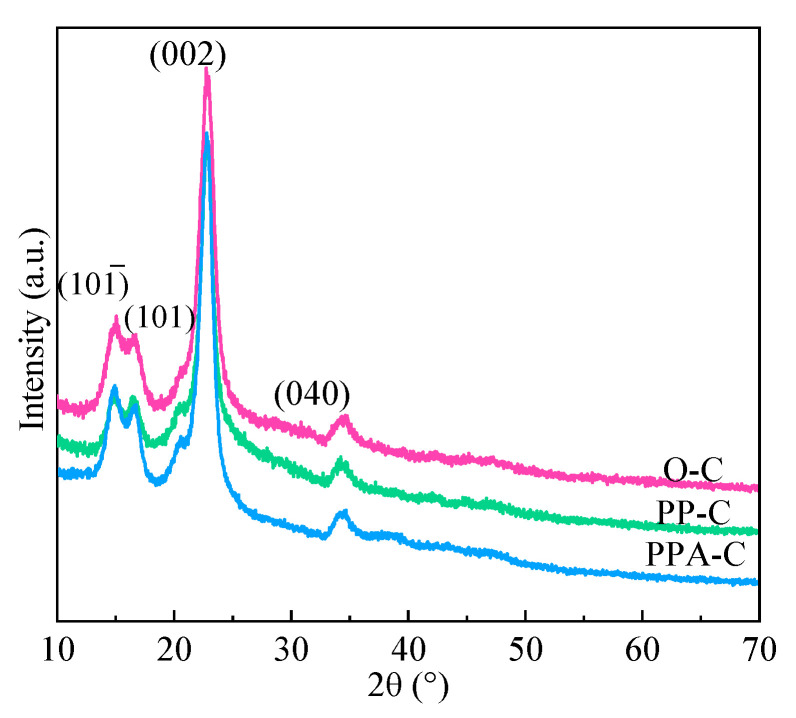
The XRD spectrum of fabrics.

**Figure 3 materials-15-02537-f003:**
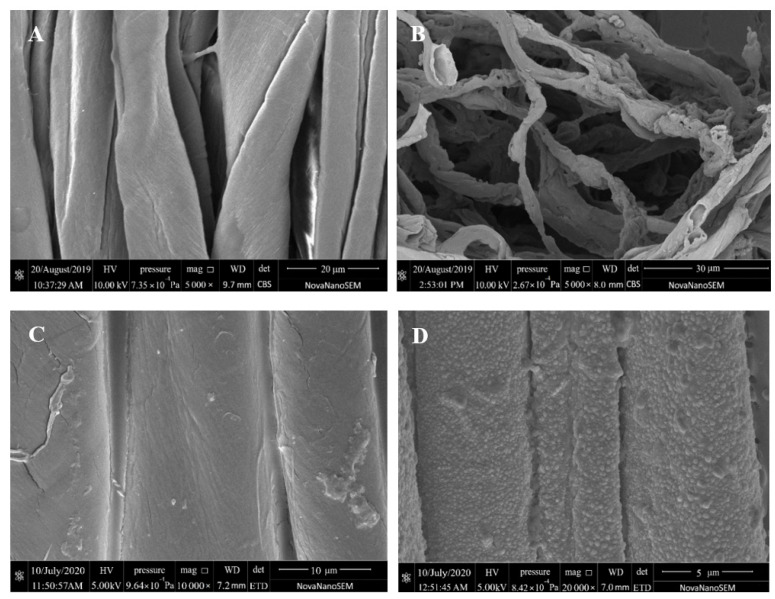
The SEM spectrogram of fabrics and TEM spectrogram of Ag. (**A**): O-C before burning; (**B**): O-C after burning; (**C**): PP-C before burning; (**D**): PP-C after burning; (**E**): PPA-C before burning; (**F**): PPA-C after burning; (**G**,**H**): different multiples of TEM.

**Figure 4 materials-15-02537-f004:**
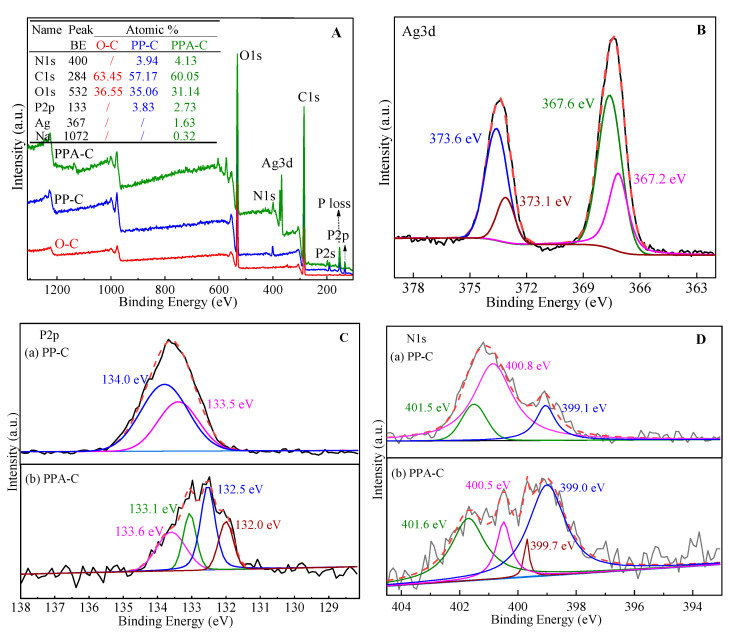
XPS spectrum of fabrics ((**A**): surface elements of fabric before and after finishing; (**B**–**D**): morphology of Ag, P, N elements on the surface of finished fabrics).

**Figure 5 materials-15-02537-f005:**
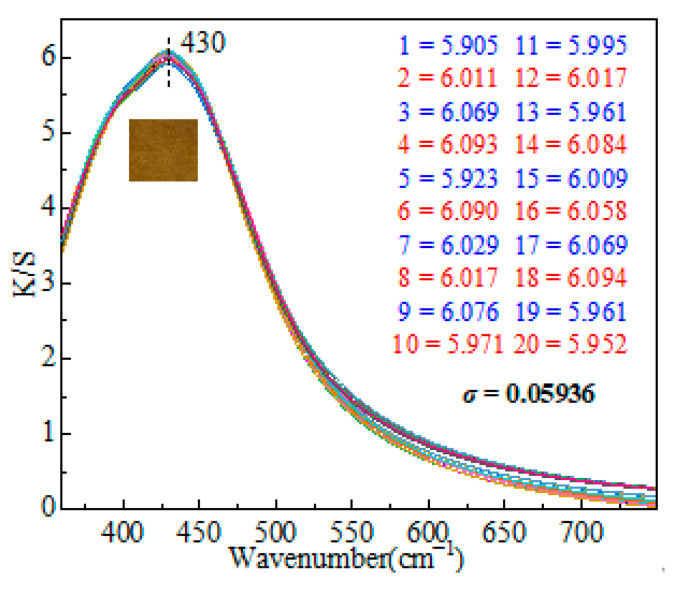
K/S spectrum of fabrics.

**Figure 6 materials-15-02537-f006:**
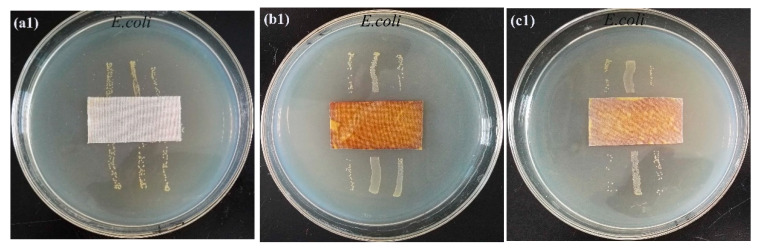
Antibacterial activities of different cotton fabrics (**a1**): O-C; (**b1**): PPA-C; (**c1**): PPA-C after washing 10 cycles with *Escherichia coli; and* (**a2**): O-C; (**b2**): PPA-C; (**c2**): PPA-C after washing 10 cycles with *Staphylococcus aureus*.

**Figure 7 materials-15-02537-f007:**
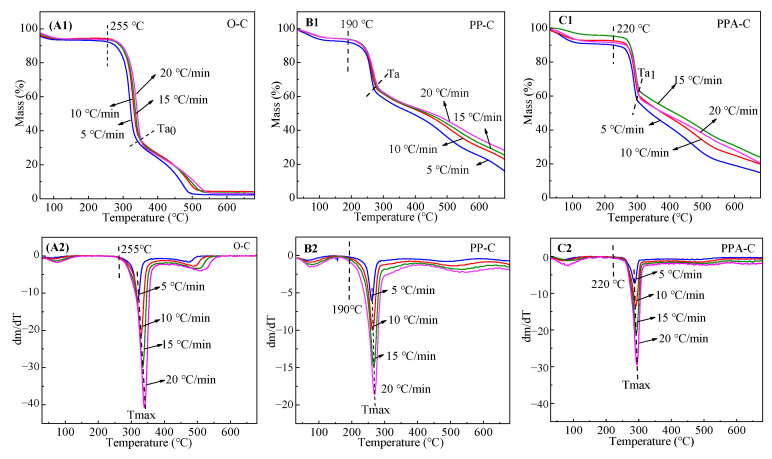
TG–DTG of cotton fabrics at different heating rates (air atmosphere). (**A1**–**C1**) were the TG of the O-C, PP-C, PPA-C fabrics respectively and (**A2**–**C2**) were the DTG of the O-C, PP-C, PPA-C fabrics respectively.

**Figure 8 materials-15-02537-f008:**
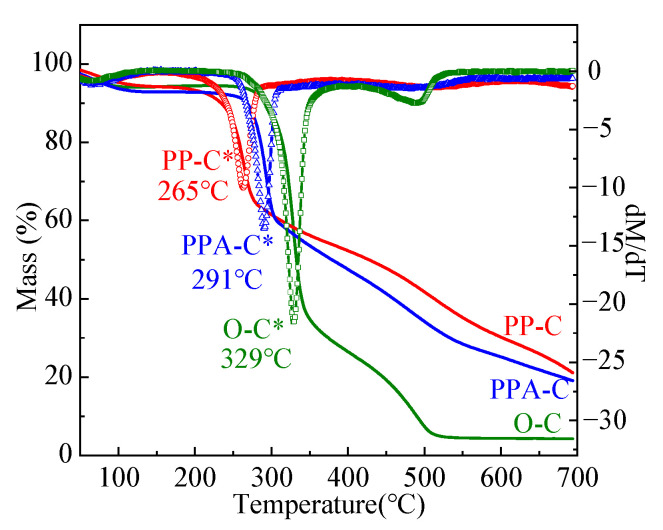
TG–DTG (*) of cotton fabrics at 10 °C/min heating rate (air atmosphere).

**Table 1 materials-15-02537-t001:** The LOI of the finishing fabrics.

Washing Times	O-C	PP-C	PPA-C
1	18	37	27
10	18	35	24
20	18	31	21

**Table 2 materials-15-02537-t002:** Antibacterial width of fabrics.

Fabrics	*T* (mm)		*W* (mm)	
	*E.coli*	*S.aureus*	*E.coli*	*S.aureus*
O-C	0	0	0	0
PPA-C	35.56	33.64	5.28	4.32
PPA-C-10	34.39	32.75	4.69	3.87

**Table 3 materials-15-02537-t003:** TG parameters of flame-retardant fabric at different heating rates.

Fabrics	Parameters	Heating Rates (°C/min)			
		5	10	15	20
O-C	Ti	234	258	258	258
	Tmax	320	329	334	340
	Ta_0_	340	350	350	350
	CR (%)	2.19	4.26	3.37	3.01
PP-C	Ti	190	190	190	190
	Tmax	258	265	269	273
	Ta	270	274	278	280
	CR (%)	14.22	21.36	23.94	26.84
PPA-C	Ti	220	220	220	220
	Tmax	286	291	294	297
	Ta_1_	299	306	306	306
	CR (%)	13.42	18.66	20.49	18.79

## Data Availability

Not applicable.

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
