# Peer review of "In Situ Synthesis of Silver Nanoparticles on Flame-Retardant Cotton Textiles Treated with Biological Phytic Acid and Antibacterial Activity"

_materials, 2022, doi:10.3390/ma15072537_

Round 1

Reviewer 1 Report

  1. How were 10% concentration Of PEI and EH and the process parameters arrived at? Give reference.
  2. How was the concentration of silver nitrate selected? (Give reference)
  3. Can be included in the discussions section to support the ideas
  4. Figure 2: peaks 220 & 311 are not visible in the figure therefore the conclusion that additional diffraction peaks were identified needs to be rectified
  5. Figure 2: X-axis is 10 -70 (2ɵ)° yet 77.5 ° is mentioned, where is it?
  6. Indicate figure 3E and F clearly
  7. Proper indication of figures 4 A, B and C.
  8. Table 1: The title of the 1st column must be in English
  9. One figure 7 combining the curves of the three samples can be added to appreciate the different properties of the samples

Reviewer 2 Report

Manuscript titled with “Antibacterial treatment and performance of bio-phytic acid-treated flame-retardant cotton textiles containing silver nanoparticles” is about flame-retardant finish using phytic acid and antibacterial treatment with silver nanoparticles (Ag NPs). Developed fabrics were evaluated for their combustion performance, thermal performance, and antibacterial capabilities. It is well organized and well written manuscript.

Comments

In this study flame-retardant and antimicrobial functionalities were combined, however need for such kind of fabric or proposed end use is missing. Main aim of the study is missing. Why we need to develop a fabric with flame retardancy and antimictobial finish?

A literature servey should be added about phytic acid and its usage as flame retardant.

I understand that the method of applying phytic acid and Ag nanoparticles is unique and novelty of the study, so it should be better emphasize this and title should be revised to emphasize the method, Because applying Ag nanoparticles as an antibacterial agent is very well known commercial application.

Again, according to suggested end use, substrate should be choosen for application. What may be the end used of 120 ±10 g/m2, twill fabric with flame retardancy and antibacterial finish? Antibacterial finish and flame retardency may be needed for such as a public wall to wall carpet. It would be better if selected fabric should be used such kind of purposes.

“The uniformity of fabric in-situ synthesis is expressed by the K/S , however color difference (deltaE) would be better to evaluate it. Authors would explain the choice of K/S.

Check Table 1. There is 次数 ??

Give some well known flame retardent fabric LOI values just for comparison for readers.

The title of the table 2 is “Antibacterial width of fabrics”  width would be properties or anything similar?

Author Response

Thank you for your affirmation. We will continue to work hard!

Reviewer 3 Report

The manuscript entitled “Antibacterial treatment and performance of bio-phytic acid-treated flame-retardant cotton textiles containing silver nanoparticles”. The paper investigated the Antibacterial potential and characterization of nanomaterials morphology. The manuscript reads well despite and is significantly original to merit the publication in the Materials journal. Accept to publish in the current form.

Author Response

(The authors gave the same response as above.)

Reviewer 4 Report

The subject of the study is interesting, but the text needs extensive revision.

  1. Reference should be added to some parts of the text: i.e. “pose a daily threat to human health; therefore, antimicrobial fabrics are gaining popularity”.
  2. Clarify the novelty of the study. Authors cited the reference number 7, what is the difference between this study and the previous published one?
  3. Regarding the antimicrobial activity tests, clarify: a) Include how the results were classified as “excellent”, include the parameters adopted (AATCC 147-2004?). Based on this, revise the conclusion, that must be supported by the results. 
  4. The manuscript needs revision on conceptual aspects of Microbiology (this point is very critical):
  5. Bacteria (e.g., fungus, germs): fungus is not a kind of bacteria. What the authors mean by germs?
  6. antibacterial finishing ingredient: “finishing ingredient” is not a usual term in Microbiology.
  7. “causing suffocation and death” (of the microorganisms) – these terms and concepts are not correct. Please, revise.
  8. “to cool and coagulate.” (material and methods, item 2.5, when describing the preparation of culture medium) – Coagulation is not properly applied here. Please, revise accordingly.
  9. Regarding the material

The authors reported that “flame retardant textiles were submerged in a 0.1 mol/L silver nitrate solution at room temperature (bath ratio of 1:30).”. What is the stability of the silver nitrate on the surface of the material? Are the levels of silver nitrate liberated from the material safe for humans?

  1. Methodology

Include the parameters of antimicrobial activity classification according to AATCC 147-2004.

Reviewer 5 Report

The manuscript titled: "Antibacterial treatment and performance of bio-phytic acid-treated flame-retardant cotton textiles containing silver nanoparticles" proposes a method to prepare flame-retardant and antibacterial fabrics by employing phytic acid and silver nanoparticles. The manuscript is well structured and the results are quite well presented, however, there are some improvements that are needed to be made:

  1. There are a couple of unexplained abbreviations in the manuscript, namely: chapter 2.1 EH, chapter 3.1 EDTA
  2. Several figures are not marked well. Fig. 3 has marked parts A1-C2 and then some unmarked parts. Fig. 4 is marked A,B,B,B, in text this is described as Fig. 4 A-C and it should be marked as A,B,C,D and cited in the text appropriately. Consider marking Fig. 7 consistently with the previous figures as A1-C2.
  3. Table 1 has a header in chinese, please change this to english. Header in Table 3 is wrongly assigned/broken. The third column of the header should probably state: Heating rates °C/min.
  4. The assignment of Ag bands in XRD spectrum is very dubious. This is expected, since very little Ag is present in the sample compared to the cellulose content, and moreover, it is nanostructured, which further decreases the XRD signal. The presence of Ag is quite clear from the SEM and XPS, so the XRD spectrum does not need to be marked for Ag in the noise.
  5. TEM and probably HRTEM analysis was obviously performed on the AgNPs (Fig. 3 inset), however the description of theses results is missing in the text. Please, either comment on the TEM results in the manuscript, or delete the TEM images.
  6. It would improve the manuscript, if the authors could test the antibacterial effect on the PP-C samples as well to evaluate the inlfuence of the phytic acid treatment.
